# Structural Differences of the Semantic Network in Adolescents with Intellectual Disability

**Karin Nilsson** [1,2,*], **Lisa Palmqvist** [1,2], **Magnus Ivarsson** [1,2], **Anna Levén** [1,2], **Henrik Danielsson** [1,2], **Marie Annell** [1,2], **Daniel Schöld** [1,2] and **Michaela Socher** [1,2]

1 Department of Behavioural Sciences and Learning, Linköping University, 58183 Linköping, Sweden; lisa.palmqvist@liu.se (L.P.); magnus.ivarsson@liu.se (M.I.); anna.leven@liu.se (A.L.); henrik.danielsson@liu.se (H.D.); marie.e.annell@gmail.com (M.A.); daniel.schold@liu.se (D.S.); michaela.socher@liu.se (M.S.)

2 Swedish Institute for Disability Research, Linköping University, 58183 Linköping, Sweden

* Correspondence: karin.a.nilsson@liu.se

**Abstract:** The semantic network structure is a core aspect of the mental lexicon and is, therefore, a key to understanding language development processes. This study investigated the structure of the semantic network of adolescents with intellectual disability (ID) and children with typical development (TD) using network analysis. The semantic networks of the participants ($n_{ID}$ = 66; $n_{TD}$ = 49) were estimated from the semantic verbal fluency task with the pathfinder method. The groups were matched on the number of produced words. The average shortest path length (ASPL), the clustering coefficient (CC), and the network's modularity (Q) of the two groups were compared. A significantly smaller ASPL and Q and a significantly higher CC were found for the adolescents with ID in comparison with the children with TD. Reasons for this might be differences in the language environment and differences in cognitive skills. The quality and quantity of the language input might differ for adolescents with ID due to differences in school curricula and because persons with ID tend to engage in different out-of-school activities compared to TD peers. Future studies should investigate the influence of different language environments on the language development of persons with ID.

**Keywords:** semantic network analysis; intellectual disability; adolescents

## 1. Introduction

The semantic network structure is a core aspect of the mental lexicon [1] and is, therefore, a key to understanding language development processes. Different methods have been applied to study the semantic network structure in various populations in recent years [2–5]. However, little is known about the semantic network structure in persons with intellectual disability (ID), although language limitations [6], including semantic verbal fluency deficits [7,8], are part of the ID symptomatology. A better understanding of the specific characteristics of the semantic networks in persons with ID can be an essential tool for the development of language interventions for the group. It may also give important clues about semantic network development in general by shedding light on the role of general intellectual functioning. The current study aimed to investigate if the semantic network structure in a sample of adolescents with ID and a control group of younger typically developing (TD) children, differs. Studying the structure of the semantic network may lead to important insight into the verbal profile of persons with ID. Differences in the structure could help to explain specific challenges seen in language ability [9] and memory [10] in the population with ID. Such knowledge could, in the long-term, lay the foundation for the development of more effective interventions aimed at strengthening different verbal abilities based on specific network features of the ID population.

Semantic network analysis builds on graph theory and offers new ways for analyzing how information such as words generated in verbal fluency tasks are stored in memory and later retrieved [11]. The basic elements of the semantic network are nodes (words) and edges (the relationships between the words). The edges represent the associative strength between words [12]. Words that are named in temporal proximity to each other are likely to be stored nearby in the mental space [13]. Data for the network analysis are often attained through a semantic fluency task, typically involving participants naming as many words as possible within a given category and time [14].

Different characteristics of semantic networks have been studied, including distances between nodes and the tendency and nature of the cluster formation. The shortest path length is defined as the minimum number of edges (steps) between two nodes. The average shortest path length (ASPL) is the average number of edges in the shortest path between all possible pairs of nodes [12]. A high ASPL indicates that the nodes are, on average, remotely connected in the semantic network. The clustering coefficient (CC) measures the extent to which the nodes and their neighboring nodes are interconnected [12,13]. A high CC indicates that the semantic network is densely clustered. Another common quantifier of a semantic network is the network's modularity (Q). Modularity is a measure of the tendency to form subgroups (communities) within the network [12,15]. A high Q indicates well-defined subgroups with many edges connecting nodes within the subgroups and few edges between nodes belonging to different subgroups [15]. Taken together, these three measures—ASPL, CC, and Q—describe the mental representation of the semantic network in an individual's long-term memory. See Figure 1 for visual representation of the three measures.

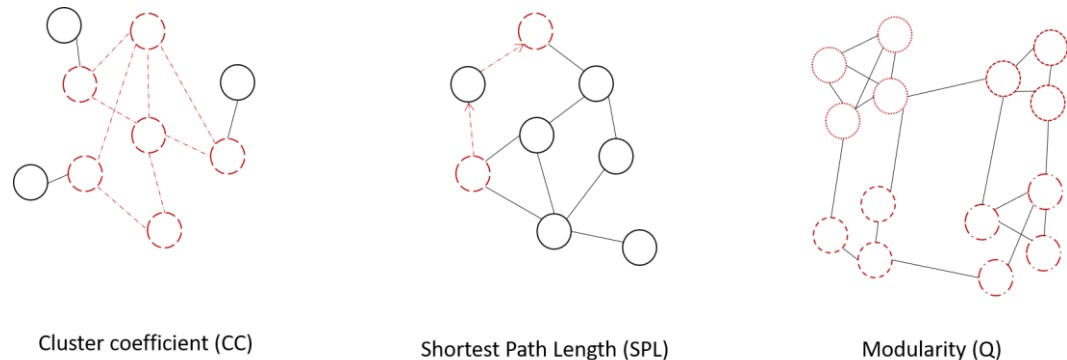

**Figure 1.** Visualization of the network measures used in the current study. The cluster coefficient measures the extent to which the nodes are interconnected. The shortest path length is the minimum number of edges between two nodes. The average shortest path length (ASPL) was calculated and used as a measure. Modularity measures the tendency to form subgroups within the network.

It has been argued that the structure of the semantic network might be the result of statistical learning, a process where taxonomic categories are formed based on co-occurrence regularities [16]. Evidence suggests that statistical learning is apparent in children as young as 4–5 years of age [16]. A prerequisite for statistical learning is that the similarities between contexts are detected and understood by the child, i.e., an ability to implicitly match patterns. Research studying statistical learning in persons with ID is sparse (see Saffran [17]), but it has been suggested that the capacity of implicit learning is functionally equivalent in young adults with and without ID [18] as well as in children and adolescents with and without ID matched on mental or chronological age [19]. However, Kover [20] argues that persons with ID may exhibit difficulties in implementing learning from distributional cues (i.e., patterns in input) and that weaker cognitive and linguistic skills may hinder efficient learning from cues. In addition, Thiessen et al. [21] suggest that the outcome of statistical learning changes during development as a function of experience and the maturity of the learner. Thus, it would be reasonable to assume that the ID

population differs from the TD population in terms of the outcome of statistical learning. If this is the case, and if statistical learning influences the structure of semantic networks, it follows that the semantic networks of persons with ID differ from those of TD peers. For example, semantic categories may be structured according to different principles in the ID and the TD groups. A specific word, such as "dog", could activate the category "house animals" in the TD group while activating random animals, words from other categories, or no other words at all in the ID group.

Statistical learning is likely not the only factor influencing the semantic network. In a conceptual framework for understanding the aging mental lexicon presented by Wulff et al. [1], learning processes are placed alongside aspects of the environment as factors that may affect the network structure. When it comes to environmental factors, Wulff et al. [1] suggest both qualitative (content) and quantitative (total amount of exposure) aspects that may be of importance. It might be the case that the environment differs between students with and without ID since the former group follows a different curriculum in Sweden [22] and tends to attend different out-of-school activities [23]. One aspect of learning highlighted by Wulff et al. [1] is that the encoding of new information is moderated by prior knowledge.

No previous study has applied network analysis to compare the semantic network structure of adolescents with ID to a TD sample based on data from a semantic fluency task. The current study will begin to fill this research gap. The number of words included in the network might influence the structure [24]. Therefore, controlling for size is essential. In the current study, this bias was reduced through the matching of groups based on the number of produced words on the semantic fluency task. The study aimed to answer the following research question: Does the structure of the semantic networks differ between adolescents with ID and children with TD, and if so, how?

Since prior research within the field of network structure in ID is scarce, there is no clear basis for formulating specific hypotheses, which motivates the explorative design of the present study. However, prior research and theories on statistical learning and semantic network development indicate the following interpretations of possible outcomes:

1. The chronological age of the ID group is higher compared with the TD group. Therefore, they should have been exposed to more language input, and their semantic network should be more developed than the semantic network of the comparison group, even if their total number of produced words are the same.
2. However, the limitations in cognitive functions might lead to the ID group not being able to make the same use of the language input as a TD group. Therefore, their semantic network might have a similar or less developed structure than the one of the comparison group.

## 2. Materials and Methods

The current study was an empirical study investigating differences in the semantic network between adolescents with ID and children with TD. Data from two different projects were used, and a network analysis using the pathfinder method was implemented. In the following sections, the sample, procedure, materials, and network analysis are described.

### 2.1. Participants and Recruitment

The verbal fluency data used in this paper are based on existing data from two different projects [25,26]. The data includes 49 participants with TD and 66 participants with ID. The participants with TD attended preschool class (i.e., Swedish school preparation class for children ~age 6). According to the teacher report, none of the children in the TD group had a developmental disability. The participants with ID were all adolescents, had an ID with unknown etiology, and attended compulsory school for students with ID. Participants in the ID group with additional disabilities, as indicated by a parental report, were excluded from the study.

Caregivers of all participants, and adolescents with ID over the age of 15, signed an informed consent form. Caregivers and participants were told they could drop out of the study without giving a reason. The data collection for the children with TD (Ref: 2015/308-31) and adolescents with ID (Ref: 2017/139-31) was approved by the local Research Ethics Review Committee in Linköping. The results from the semantic verbal fluency test of the TD group have been reported in another study about pragmatic language ability [27]. The result from the semantic verbal fluency data of the ID group is used in two pre-registered studies investigating reading ability in adolescents with ID [26].

### 2.2. Matching Procedure

The number of words included in the network might influence the structure [24]. Therefore, controlling for size is essential. This bias was reduced through the matching of groups based on the number of produced words on the semantic fluency task. Since the ID group was larger than the TD group, a subset of the ID group was selected for matching. One child from the TD group was excluded as the testing was disturbed several times. In addition, one outlier with a much higher score than the others from the TD group was excluded. Several, but not all, participants in the ID group could be individually matched to a child in the TD group. To minimize the effect of selection bias, ten different (but overlapping) samples were selected from 1,000,000 randomly generated possible selections from the ID group. These ten samples were chosen since they deviated the least from the mean (12.00) and standard deviation (4.08) of the number of produced animals in the TD group. All ten ID samples were matched on mean and SD to the TD group on the number of produced animals ($p > 0.65$; all means = 12.00, SD = 4.08 (range 4.07–4.09)). Both the TD group and the 10 different ID groups contained 47 participants each (64 different ID participants belonged to at least one selected ID group). The samples were comparable in terms of gender (21 females in the TD group; 22.5 females across the ID groups (range 20–25)). The TD group (*M* = 6:6 years, *SD* = 3.9 months) had a lower chronological age than the ID group (*M* = 15:11 years, *SD* = 27.6 months). All analyses were performed on all ten datasets. The results were pooled across datasets, and the pooled results are presented.

### 2.3. Semantic Verbal Fluency Test

The semantic network was assessed using the animal category subtest from the Delis-Kaplan Executive Function System (D-KEFS; [14]). The participants were asked to verbalize as many animals as possible in one minute. The total number of correctly generated words was used to match the groups. Fictional animals or duplicates were marked as invalid responses.

### 2.4. Procedure

The testing was conducted one-to-one in a quiet room at school (both groups) or at home (for some of the children in the TD group). The test administrator was a speech and language pathologist (both groups), an experienced test leader with a background in education (ID group), or a researcher with a background in cognitive science (TD group). All test sessions were recorded, and the recordings were used to transcribe the answers for the semantic verbal fluency task. The testing was part of two larger research projects. The children with TD as well as the adolescents with ID were tested on more tasks than are reported here.

### 2.5. Network Analysis

Only responses produced by at least two participants were included in the network analysis. We estimated the networks using the pathfinder method (see [13] for the recommendation, see [28,29] for method). The pathfinder method has been recommended for the estimation of group networks that are connected using every response, and networks are based on edge similarity [13]. The topological properties of the networks were validated using a bootstrapped random network analysis [24]. To compare the network of the ID

group and the TD group, a case-drop bootstrap analysis with 2000 runs was performed. This analysis was performed with all 10 datasets. ASPL, CC, and Q were thereafter pooled. Because the visualization of the networks cannot be pooled, the comparison was done for all 10 datasets with ID that gave the same results. To improve the readability of the results section, only one ID network is presented (for the dataset closest to the median values on a combination of ASPL, CC, and Q).

All analyses were performed in R [30] using the R packages tidyverse [31], readxl [32], tictoc [33], beepr [34], stringr [35], flextable [36], SemNetCleaner, SemNetDictionaries, and SemNeT [37]. An alpha level of 0.05 was used.

## 3. Results

### 3.1. Number of Unique Words

The adolescents with ID produced 129 (pooled result: 128.9) unique animals. Of these animals, 43.6 (39.8%) were only produced by one child. The children with TD produced 106 unique animals. Of these animals, 43 (40.6%) were produced by only one child.

### 3.2. Network Validation

ASPL, CC, and Q of all networks differed significantly from random ($p < 0.001$). The results of the random comparison are reported in Appendix A.

### 3.3. Network Comparison

The results from the pooled bootstrap analyses are reported in Table 1. A significantly smaller ASPL and Q were found for the adolescents with ID in comparison with the children with TD. In addition, a significantly higher CC was found for the adolescents with ID in comparison with the children with TD.

**Table 1.** The result from the pooled bootstrap analyses for ASPL, CC, and Q.

|  | ID Mean | TD Mean | $F(1,1997)$ | $p$ | Partial $\eta2$ | Direction |
|---|---|---|---|---|---|---|
| **ASPL** | 2.73 | 2.84 | 178 | <0.001 | 0.08 | ID < TD |
| **CC** | 0.48 | 0.43 | 800 | <0.001 | 0.27 | ID > TD |
| **Q** | 0.35 | 0.37 | 308 | <0.001 | 0.12 | ID < TD |

As can be seen in Figures 2 and 3, the network of the adolescents with ID includes more close nodes and exhibits a shorter ASPL. In addition, the network is less spread out. Neither the network of the children with TD nor the network of the adolescents with ID appear to have clear subgroups. Rather, many words are not clearly separated. For the network of the children with TD, there is a tendency towards the development of subgroups, which is not the case as much for the network of the adolescents with ID. This is also mirrored in the Q, which is significantly larger for the TD group compared with the ID group. Further, the adolescents with ID exhibit a higher CC compared with the children with TD. This is visible in the figures, as the largest subgroup of the ID group is more densely clustered (see bottom left in Figure 2) compared with the largest subgroup of the TD group (see top right in Figure 3). For the TD group, the developing subgroups are mostly related to the expected taxonomic structure of the animal category, while this is true to a lesser extent for the ID group. Note that the network layout was created using the Fruchterman–Reingold algorithm [37], which is very sensitive to small differences in network properties such as path length. The position of the large subgroups on the opposite ends for the ID group compared with the TD group is therefore merely an artifact of how the network plot was created.

ID

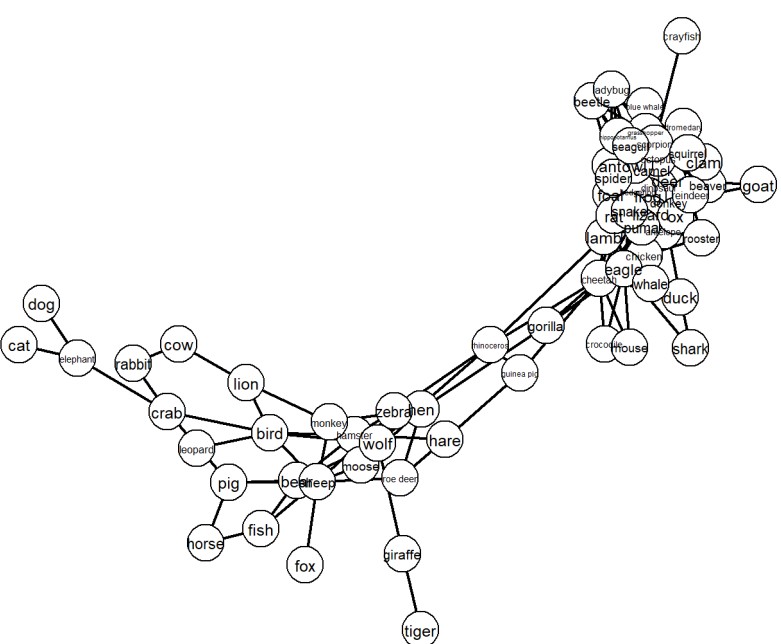

**Figure 2.** Graphical representation of the semantic network of the adolescents with ID.

TD

**Figure 3.** Graphical representation of the semantic network of the children with TD.

## 4. Discussion

The current study compared the semantic network structure in a group of adolescents with ID and a group of younger children with TD, matched on the produced number of words on a semantic fluency task. This is, to our knowledge, the first attempt to quantify the semantic network in the ID population. The results showed that the structure of the semantic networks differs between the groups. The semantic network of the adolescents with ID has a significantly smaller ASPL and Q and a significantly larger CC compared to with the semantic network of the children with TD. Adolescents with ID in this study

have a more condensed semantic network structure compared with children with TD, which indicates that the semantic network for the adolescents with ID is less developed. Similar results have been found for children with cochlear implants (CI; [24]) and second-language speakers [3]. Kenett et al. [24] compared a group of children with CI with a group of age-matched, typical-hearing peers. The CI group had a significantly smaller ASPL compared with the typical-hearing group. Kenett et al. [24] interpreted this result as the CI group having a less developed semantic network structure. Borodkin et al. [3] showed that second language speakers had a lexical network with a larger CC and a smaller Q in comparison with its first language equivalent. This result was interpreted as the second language speakers' network being less well-organized, as the words in the network were less likely to be grouped into identifiable subcategories [3]. Similar to the findings by Borodkin et al. [3], the current study found a lower Q value in the ID group compared with the TD group, indicating a less developed taxonomic structure of the semantic network in adolescents with ID.

Wulff et al. [1] proposed a framework for understanding the mechanisms behind age differences in the mental lexicon. We suggest that the components of this framework can be used in explaining the less developed semantic network of adolescents with ID. Wulff et al. [1] argue that the environment plays an important role in the structure of the semantic lexicon. It could be that the quality and/or quantity of the language input differs between adolescents with ID and children with TD.

The adolescents in the current study were all enrolled in special schools, meaning that they were exposed to a different learning environment compared to the children with TD. The special schools in Sweden follow a different curriculum [22]. This curriculum also provides more opportunities for individual adaptations of teaching [22], meaning that the learning environment might vary between students enrolled at the same special school. The language input for the adolescents with ID can therefore be assumed to be heterogeneous, which in turn means that a greater variation in verbal fluency performance can be expected. This could be a contributing factor as to why the estimated semantic network is less structured. Similar reasoning was used by Borodkin et al. [3], who argued that a possible explanation for the less well-organized semantic network in second language speakers could be the heterogeneous language proficiency in that group.

There has been some evidence that adolescents with ID engage in different out-of-school activities compared to their typically developing peers [23]. The difference in educational and out-of-school environments may affect the quality and/or the quantity of the linguistic input. In addition, it has been shown that parents of children with a delayed language development tend to adjust their language level on several quality measures [38], and in line with the reasoning of Beckage et al. [39], this could create a linguistic environment with different structural properties compared to the TD group.

Wulff et al. [1] proposed learning as another component that is vital for the mental lexicon. As laid out in the Introduction, statistical learning is of importance for the development of the semantic network (see: [16]). A less developed semantic lexicon for adolescents with ID could be explained by reduced statistical learning ability. In addition, an important aspect of learning is prior knowledge, meaning that the encoding of new information is moderated by pre-existing knowledge [1]. Studies have shown that the level of acquired language predicts further learning from distributional cues in infants [40,41], and suggestions have been made that the delayed language development may constrain the usage of cues [21]. Kover [20] argues that even if persons with ID may have more experience as measured by chronological time, the knowledge might be less accumulated due to poorer learning efficiency.

Currently, little is known about the effects that the structure of the semantic network has on the higher-order language ability of adolescents with ID. A less structured semantic network likely makes language understanding and production more demanding, as words might not be activated automatically (or the wrong ones might be activated). This is in

accordance with studies showing that a shorter ASPL and a higher CC might make it harder to identify words and might lead to confusing words in memory [42,43].

To conclude, adolescents with ID have a less structured semantic network than children with TD even when the network size is controlled for. These differences might be due to differences in the language environment as well as to differences in cognitive skills. If the language environment is an important factor for the structure of the semantic network of persons with ID, interventions should aim to increase the quality and quantity of the language input that children and adolescents with ID receive. The less structured semantic network might be an important underlying factor for language problems in persons with ID.

### 4.1. Future Studies

This is a novel field of research, and more studies are needed to disentangle the effects of different factors on the semantic network structure in persons with ID. One way of differentiating the effect of cognitive ability and the effect of the language input could be cognitive modeling. A simulation study using a semantic network model could help to investigate which type of behavior a network would display with less qualitative language input and which behavior it would display with reduced statistical learning ability. This kind of study could also help to investigate how the structure of the semantic network is influencing language ability in persons with ID. The magnitude of the differences in the current study was small (cf. [24]), and it is currently not known if these small differences in the structure influence real-life language abilities. In addition, more studies are needed to investigate the effects of different learning environments and their relation to the quality and quantity of language input.

### 4.2. Limitations

This study was conducted using data from two different research projects. A coordinated data collection would have allowed the research team to collect more data on related linguistic and cognitive abilities. The sample size in this study should be considered large concerning the tradition within disability research. However, when estimating networks, a larger sample size would be desirable to make sure the estimated networks are stable.

**Author Contributions:** Conceptualization, K.N., M.S., L.P., M.I., A.L., H.D., M.A. and D.S.; methodology, M.S., L.P. and H.D.; software, M.S., L.P. and H.D.; validation, M.S.; formal analysis, M.S., L.P. and H.D.; investigation, K.N., M.A. and M.S.; data curation, M.A. and L.P.; writing—original draft preparation, K.N., M.S., L.P., M.I., A.L., H.D., M.A. and D.S.; writing—review and editing, K.N., M.S., L.P., M.I., A.L., H.D., M.A. and D.S.; visualization, M.S.; administration, K.N., M.S. and L.P.; funding acquisition, H.D. All authors have read and agreed to the published version of the manuscript.

**Funding:** This research was funded by European Union Seventh Framework Program (FP7/2007–2013) under Grant Agreement FP7-607139 (iCARE) and by the Swedish Research Council (2013-01363 and 2016-04217).

**Institutional Review Board Statement:** The study was conducted according to the guidelines of the Declaration of Helsinki, and approved by the regional Research Ethics Committee in Linköping, Sweden (2017/139-31 and 2015/308-31).

**Informed Consent Statement:** Informed consent was obtained from all subjects involved in the study.

**Data Availability Statement:** The data on the children with TD presented in this study are available on request from the corresponding author. The data are not publicly available due to restrictions in the ethics application. The data for the adolescents with ID are available on https://osf.io/aet7b/ (accessed on 23 April 2021).

**Acknowledgments:** We would like to thank Alexander P. Christensen for the help and support with the SemNet packages in R.

**Conflicts of Interest:** The authors declare no conflict of interest.

## Appendix A

**Table A1.** Random Network comparison for the network of the children with TD, $p < 0.001$ for all comparisons.

|  | TD | M | SD |
|---|---|---|---|
| ASPL | 3.47 | 2.25 | 0.03 |
| CC | 0.38 | 0.28 | 0.02 |
| Q | 0.35 | 0.23 | 0.01 |

**Table A2.** Random Network comparison for the network of the adolescents with ID (dataset 1), $p < 0.001$ for all comparisons.

|  | ID | M | SD |
|---|---|---|---|
| ASPL | 2.73 | 2.03 | 0.02 |
| CC | 0.45 | 0.37 | 0.02 |
| Q | 0.31 | 0.17 | 0.01 |

**Table A3.** Random Network comparison for the network of the adolescents with ID (dataset 2), $p < 0.001$ for all comparisons.

|  | ID | M | SD |
|---|---|---|---|
| ASPL | 2.82 | 2.08 | 0.02 |
| CC | 0.45 | 0.34 | 0.02 |
| Q | 0.33 | 0.19 | 0.01 |

**Table A4.** Random Network comparison for the network of the adolescents with ID (dataset 3), $p < 0.001$ for all comparisons.

|  | ID | M | SD |
|---|---|---|---|
| ASPL | 3.23 | 2.09 | 0.02 |
| CC | 0.47 | 0.38 | 0.02 |
| Q | 0.33 | 0.19 | 0.01 |

**Table A5.** Random Network comparison for the network of the adolescents with ID (dataset 4), $p < 0.001$ for all comparisons.

|  | ID | M | SD |
|---|---|---|---|
| ASPL | 2.62 | 2.01 | 0.01 |
| CC | 0.47 | 0.31 | 0.01 |
| Q | 0.36 | 0.18 | 0.01 |

**Table A6.** Random Network comparison for the network of the adolescents with ID (dataset 5), $p < 0.001$ for all comparisons.

|  | ID | M | SD |
|---|---|---|---|
| ASPL | 2.70 | 1.97 | 0.01 |
| CC | 0.50 | 0.45 | 0.02 |
| Q | 0.28 | 0.15 | 0.01 |

**Table A7.** Random Network comparison for the network of the adolescents with ID (dataset 6), $p < 0.001$ for all comparisons.

|  | ID | M | SD |
|---|---|---|---|
| ASPL | 2.79 | 1.98 | 0.01 |
| CC | 0.50 | 0.45 | 0.02 |
| Q | 0.23 | 0.14 | 0.01 |

**Table A8.** Random Network comparison for the network of the adolescents with ID (dataset 7), $p < 0.001$ for all comparisons.

|  | ID | M | SD |
|---|---|---|---|
| ASPL | 3.12 | 2.1305 | 0.0177 |
| CC | 0.44 | 0.31 | 0.02 |
| Q | 0.33 | 0.20 | 0.01 |

**Table A9.** Random Network comparison for the network of the adolescents with ID (dataset 8), $p < 0.001$ for all comparisons.

|  | ID | M | SD |
|---|---|---|---|
| ASPL | 2.98 | 2.05 | 0.02 |
| CC | 0.46 | 0.34 | 0.02 |
| Q | 0.32 | 0.18 | 0.01 |

**Table A10.** Random Network comparison for the network of the adolescents with ID (dataset 9), $p < 0.001$ for all comparisons.

|  | ID | M | SD |
|---|---|---|---|
| ASPL | 3.12 | 2.04 | 0.02 |
| CC | 0.46 | 0.38 | 0.02 |
| Q | 0.30 | 0.17 | 0.01 |

**Table A11.** Random Network comparison for the network of the adolescents with ID (dataset 10), $p < 0.001$ for all comparisons.

|  | ID | M | SD |
|---|---|---|---|
| ASPL | 2.57 | 2.08 | 0.02 |
| CC | 0.41 | 0.32 | 0.02 |
| Q | 0.30 | 0.19 | 0.01 |

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
