# Peer review of "Structural Differences of the Semantic Network in Adolescents with Intellectual Disability"

_2504-2289, doi:10.3390/bdcc5020025_

Round 1
Reviewer 1 Report
- - P. 2, lines 47 - 60: It would be advisable to illustrate each of the three key measures with simple graphical examples.
- The research question is mentioned on p. 3, l. 105. In my view, it appears too late; I would like to see it at the very beginning of the article.
- Although the language of the article is by no means bad, it would still require a check / proofreading by a native speaker; e.g. there should be a comma on p. 3, l. 107 after scarce, and there are some issues on p. 3, l. 131 - 132 :
The caregivers of the participants in both groups and by the participants...
All the participants and their caregivers...
This just shows the need for one more language check; I did not revise the entire text.
- Explain Figures 1 and 2 more in detail to the reader: now the reader easily gets the impression that there is (almost) a similar cluster also in the results for TD. Also explain the reader why cluster formation takes place at opposite ends between the Figures. You should also discuss the lexical meaning of the elements in the Figures, and its connection with the structure of the Figures.
Author Response
Thank you for your valuable comments on our manuscript. We agree with your suggestions, and we outline below what changes that have been made to the manuscript.
- P. 2, lines 47 - 60: It would be advisable to illustrate each of the three key measures with simple graphical examples.
- We agree that using graphical examples makes the method more transparent to the reader. We have added a figure explaining our key measures after this paragraph (p. 2).
The research question is mentioned on p. 3, l. 105. In my view, it appears too late; I would like to see it at the very beginning of the article.
- We have rephrased our aim with the study (p.1 l. 35-38), so that the research question becomes clearer in the first paragraph of the introduction.
Although the language of the article is by no means bad, it would still require a check / proofreading by a native speaker; e.g. there should be a comma on p. 3, l. 107 after scarce, and there are some issues on p. 3, l. 131 - 132 :The caregivers of the participants in both groups and by the participants... All the participants and their caregivers...This just shows the need for one more language check; I did not revise the entire text
- A native speaker has proofread our manuscript and we have made all changes suggested by the proofreader.
Explain Figures 1 and 2 more in detail to the reader: now the reader easily gets the impression that there is (almost) a similar cluster also in the results for TD. Also explain the reader why cluster formation takes place at opposite ends between the Figures. You should also discuss the lexical meaning of the elements in the Figures, and its connection with the structure of the Figures.
- We have explained the figures in more detail in the network comparison section (p. 6 l. 238-255). The network layouts have been created using the Fruchterman-Reingold algorithm (Christensen & Kenett, 2019) which is very sensitive to small differences in network properties such as path length. The difference of the placement of the large clusters is therefore not meaningfull, but merely an artifact of how the networks are plotted. We could have created graphs with matching layouts but that would mean we would fix the nodes at specific places, which would have led to misleading graphs as the developing subgroups would not have been plotted correctly for the two different groups. We have added a short explanation about this as well.
Reviewer 2 Report
This is a well-written article. It is original and the results are well presented. My suggestion is to state explicitly the scope and content of your study.

Author Response
Thank you for your valuable comments on our manuscript. We agree with your suggestions, and we outline below what changes that have been made to the manuscript.
- after line 38: draw some implications
-
- We have moved a paragraph that describes the implications to the beginning of the introduction (p. 1 l. 38-44)
- after line 67: explain how this is relevant to your study
-
- When re-reading the manuscript we realized that we do not think the usage-based models are important for our study so we omitted the sentence. Further, we merged two paragraphs (p. 3 l. 83) in order to make the importance of statistical learning clearer.
- before section 2.1: add a section entitled as “scope & content”
-
- We rephrased the first paragraph of the study to make sure that the scope and aim are clear from the beginning (p.1 l. 35-38). In addition, we added a description of the overall content of the methods before section 2.1. (p. 4, l. 137-141).